# DEEP REINFORCEMENT LEARNING FOR WIRELESS SCHEDULING WITH MULTICLASS SERVICES

## ABSTRACT

In this paper, we investigate the problem of scheduling and resource allocation over a time varying set of clients with heterogeneous demands. This problem appears when service providers need to serve traffic generated by users with different classes of requirements. We thus have to allocate bandwidth resources over time to efficiently satisfy these demands within a limited time horizon. This is a highly intricate problem and solutions may involve tools stemming from diverse fields like combinatorics and optimization. Recent work has successfully proposed Deep Reinforcement Learning (DRL) solutions, although not yet for heterogeneous user traffic. We propose a deep deterministic policy gradient algorithm combining state of the art techniques, namely Distributional RL and Deep Sets, to train a model for heterogeneous traffic scheduling. We test on diverse number scenarios with different time dependence dynamics, users' requirements, and resources available, demonstrating consistent results. We evaluate the algorithm on a wireless communication setting and show significant gains against state-of-the-art conventional algorithms from combinatorics and optimization (e.g. Knapsack, Integer Linear Programming, Frank-Wolfe).

## 1 INTRODUCTION

User scheduling (i.e., which user to be served when) and associated resource allocation (i.e., which and how many resources should be assigned to scheduled users) are two long-standing fundamental problems in communications, which have recently attracted vivid attention in the context of next generation communication systems (5G and beyond). The main reason is the heterogeneity in users' traffic and the diverse Quality of Service (QoS) requirements required by the users. The goal of this paper is to design a scheduler and resource assigner, which takes as inputs the specific constraints of the traffic/service class each user belongs in order to maximize the number of satisfied users.

This problem is hard to solve since we have at least two main technical challenges: (i) except for some special cases, there is no simple closed-form expression for the problem and *a fortiori* for its solution; (ii) the problem solving algorithm has to be scalable with the number of users. Current solutions rely on combinatorial approaches or suboptimal solutions, which seem to work satisfactorily in specific scenarios, failing though to perform well when the number of active users is large. This motivates the quest for alternative solutions; we propose to resort to Deep Reinforcement Learning (DRL) to tackle this problem.

In the context of DRL, we propose to combine together several ingredients in order to solve the aforementioned challgening problem. In particular, we leverage on the theory of Deep Sets to design permutation equivariant and invariant models, which solves the scalability issue, i.e., the number of users can be increased without having to increase the number of parameters. We also stabilize the learning process by adding in a new way the distributional dimension marrying it with Dueling Networks to "center the losses".

Finally, we compare the proposed DRL-based algorithm with conventional solutions based on combinatorial or suboptimal optimization approaches. Our experiments and simulation results clearly show that our DRL method significanlty outperforms conventional state-of-the-art algorithms.

## 2 RELATED WORK

The scheduling problem is a well known problem appearing in various fields and as technologies progress and more people want to take advantage of the new services, how to schedule them in an efficient way becomes more intricate. This is exactly the case in wireless communication systems. Researchers are resorting to new methods, such as deep reinforcement learning, which have shown impressive results Mnih et al. (2015); Silver et al. (2016). For example in (Chinchali et al., 2018) they perform scheduling on a cellular level using Deep Reinforcement learning (DRL). Also ideas using DRL in a distributed way to perform dynamic power allocation has appeared in (Naparstek & Cohen, 2018; Nasir & Guo, 2019). Nevertheless, to the best of our knowledge, the problem of scheduling on traffic of users with heterogeneous performance requirements has not been appropriately addressed. To solve this hard problem, one can resort to distributional Reinforcement Learning researched in Jaquette (1973) and followed by (Dabney et al., 2018a;b) in order to have richer representations of the environment and obtain better solutions. Also techniques like noisy network for better explorations (Fortunato et al., 2018) or architectures like dueling networks (Wang et al., 2016) have greatly improved stability of the trained models. Finally ideas (Zaheer et al., 2017) managed to simplify and improve neural network models when permutation invariance properties apply. We combine those ideas with a deep deterministic policy gradient method (Lillicrap et al., 2016) to reach a very efficient scheduling algorithm.

## 3 THE SCHEDULING AND RESOURCE ALLOCATION PROBLEM

### 3.1 THE PROBLEM

The problem we consider here involves a set of randomly arriving users that communicate wirelessly with a base station (service provider); users require that their traffic is served according to the quality of service (QoS) requirements imposed by the service class they belong to. We consider the case where users belong to different service classes with heterogeneous requirements. Each class specifies the amount of data to be delivered, the maximum tolerable latency, and the "importance/priority" of the user. A centralized scheduler (at the base station) at each time step takes as input this time varying set of users belonging to different service classes and has to decide how to allocate its limited resources per time step in order to maximize the long-term "importance" weighted sum of satisfied users. A user is considered to be satisfied whenever it successfully received its data within the maximum tolerable latency specified by its service class.

The hard problem of scheduling and resource allocation - which is combinatorial by nature - is exacerbated by the wireless communication, which in turn brings additional uncertainty due to time-varying random connection quality. The scheduler that assigns resources does not exclude the possibility of a bad connection (low channel quality) which renders data transmission unsuccessful. In order to mitigate that effect, some protocols make use of channel state information (CSI) to the transmitter, i.e., the base station/scheduler knows in advance the channel quality and adapts the allocated resources to the instantaneous channel conditions. We consider here two extreme cases of channel knowledge: (i) *full-CSI*, in which perfect (instantaneous, error free) CSI is provided to the scheduler enabling accurate estimation of exact resources each user needs; and (ii) *no-CSI*, in which the scheduler is agnostic to the channel quality. In case of unsuccessful/erroneous data reception, we employ a simple retransmission protocol (HARQ-type I). A widely used way to model the channel connection dynamics is to make the wireless channel quality depend on the distance of the user from the base station and evolve in a Markovian way from the quality on the channel realization of the previous time step. The mathematical description of the traffic generator model and the channel dynamics is provided in the appendix A.

To better understand the problem, we draw the following *analogy*. Imagine a server having a water pitcher that is full at every time step and has to distribute it across a set of people. Every person has a glass and leaves satisfied only if its glass is filled (or overfilled) at any time instant prior to certain maximum waiting time. As mentioned before, in this work we consider a retransmission protocol (HARQ-type I), which in our analogy means that the server cannot fill a glass using multiple trials; if a glass is not filled completely at a time step, then it will be emptied and the server has to retry. The wireless communication setting brings the additional complication that the sizes of the glasses are not actually fixed but fluctuate (due to the randomness of the connection quality of each user). In

the full-CSI case, the server could know at every time step the size of glasses and therefore the exact amount of resources required. On the other hand, with no-CSI, the server can only roughly estimate the size, mainly using the amount of data requested and the distance between user and base station.

The problem can be modeled as a Markov Decision Process(MDP) (Bellman, 1957) $(\mathcal{S}, \mathcal{A}, R, P, \gamma)^1$, where $\mathcal{S}$ is the state space of the environment (described in detail in appendix A.2), and $\mathcal{A}$ is the action space (in our case the set of all feasible allocations). After action $a_t \in \mathcal{A}$ at state $s_t \in \mathcal{S}$ a reward $r_t \sim R(\cdot|s_t, a_t)$ is obtained and the next state follows the probability $s_{t+1} \sim P(\cdot|s_t, a_t)$. The discount factor is $\gamma \in [0, 1)$. Under a fixed policy $\pi : \mathcal{S} \to \mathcal{A}$ the return is a random variable defined as $Z_t^\pi = \sum_{i=0}^\infty \gamma^{t+i} r_{t+i}$ representing the discounted sum of rewards when a trajectory of states is taken following the policy $\pi$. An agent (scheduler) ideally aims to find the optimal policy $\pi^\star$ maximizing the mean reward $\mathbb{E}_\pi[Z^\pi]$. Being more rigorous, only in the full-CSI case the states $s_t$ are fully observed, whereas for no-CSI, the channel qualities are unknown to the agent and the observation $o_t \subset s_t$ is a part of the state, leading to a Partially Observable MDP (POMDP)(Åström, 1965). A way to reduce POMDP to MDP is by substituting the states with the "belief" of the states (Kaelbling et al., 1998) of the value of $s_t$. Another way is using the complete history $\{o_0, a_0, o_1, a_1, \cdots, a_{t-1}, o_{t-1}\}$, which fortunately in our case it works since only the most recent part is relevant, i.e., the one representing if and how many resources have been previously allocated to only the currently active users.

## 3.2 THE DEEP REINFORCEMENT LEARNING APPROACH

Deep reinforcement learning (DRL) has shown impressive results in many problems modeled as MDP, but mainly in cases where the environment is close to deterministic due to game rules (Atari, Chess, Go (Mnih et al., 2015; Silver et al., 2017; 2016)) or physical laws (robotics and physics tasks (Kober et al., 2013; Lillicrap et al., 2015)). A very relevant question to ask is whether we can develop a DRL algorithm coping successfully with environments exhibiting high variance randomness as in our case due to the channel dynamics and heterogeneous traffic. Existing applications of DRL to problems with similar properties in other fields are encouraging (trading, pricing, vehicle routing(Nazari et al., 2018; Charpentier et al., 2020)).

### 3.2.1 POLICY NETWORK

Our objective is to a scheduler that can handle a large number of users $K$, say $K = 100$, in which case the action space becomes infeasibly large for a traditional Deep Q-learning Network approach. For that, we propose to employ a deep deterministic policy gradient (DDPG) method (Lillicrap et al., 2016), with which we aim at training a policy $\pi_\theta : \mathcal{S} \to \mathcal{A}$ modelled as a Neural Network (NN) with parameters $\theta$. Moreover, our method should work in both full-CSI and no-CSI cases with minor, if any, modification. With full-CSI the exact amount of required resources (bandwidth) per user is known, so the (discrete) action is just to select the subset of user to satisfy, while for no-CSI, it is continuous since on top of selecting the users, the scheduler has to decide on the portion of resources each user takes. For no-CSI those portion are exactly the output of $\pi_\theta$ but for full-CSI we do a continuous relaxation[2] and the output provides the value (related to importance) per resources; that way, a user ranking is obtained, which allows the scheduler to proceed sequentially: the scheduler serves/satisfies as many of the most "valuable" (highest rank) users as possible subject to available resources. This discrepancy in the output process is the only difference in the model between full-CSI and no-CSI.

Setting $Z^\pi(s_t, a_t) = r_t + Z_{t+1}^\pi$ with $r_t \sim R(\cdot|s_t, a_t)$ being the return if at $t$ the action $a_t$ is taken followed by policy $\pi$ and $Q^\pi(s_t, a_t) = \mathbb{E}[Z^\pi(s_t, a_t)]$ be the expected return conditioned on the action at $s_t$ is $a_t$ then the objective of the agent to maximize is $J(\theta) = \mathbb{E}_{s_{t_0} \sim p_{t_0}}[Q^{\pi_\theta} s_{t_0}, \pi_\theta(s_{t_0}))]$ with $p_{t_0}$ being the probability of the initial state $s_{t_0}$ at time $t_0$. The gradient can be written (Silver et al., 2014):

$$\nabla_\theta J(\theta) = \mathbb{E}_{s_{t_0} \sim p_{t_0}, s \sim \rho_{s_{t_0}}^{\pi_\theta}} [\nabla_\theta \pi_\theta(s) \nabla_a Q^{\pi_\theta}(s, a)|a = \pi_\theta(s)] \quad (1)$$

---

[1]The only discrepancy is that the scheduler aims to ideally maximize the sum of rewards, i.e., for $\gamma = 1$, and not the discounted one.

[2]The continuous relaxation is also mandatory for a DDPG approach to work so that the gradients can pass from the value network.

with $\rho_{s_{t_0}}^{\pi_\theta}$ the discounted state (improper) distribution defined as $\rho_{s_{t_0}}^{\pi_\theta}(s) = \sum_{i=0}^\infty \gamma^i \mathbb{P}(s_{t+i} = s|s_{t_0}, \pi_\theta)$. In practice $\rho_{s_{t_0}}^{\pi_\theta}$ is approximated by the (proper) distribution $\varrho_{s_{t_0}}^{\pi_\theta}(s) := \sum_{i=0}^\infty \mathbb{P}(s_{t+i} = s|s_{t_0}, \pi_\theta)$. To compute the gradient, the function $Q^{\pi_\theta}(s, a)$ is needed which is approximated by another NN $Q_\psi(s, a)$, named value network, described in the next subsection.

We now explain the architecture of the model $\pi_\theta$. The policy falls in a category of permutation equivariant functions meaning that permuting the users should only result in permuting likewise the resource allocation. In (Zaheer et al., 2017) necessary and sufficient conditions are shown for permutation equivariance in neural networks; we adopt their model with minor changes. At first the characteristics $x_i \in \mathbb{R}^{N_x}, i \in \{1, \cdots K\}$ of each (active) user are processed individually by the same function $\phi_{user} : \mathbb{R}^{N_x} \to \mathbb{R}^{H_x}$ modeled as a two layer fully connected network. Then all those features per user are aggregated with the permutation equivariant $f_\sigma : \mathbb{R}^{K \times H} \to \mathbb{R}^{K \times H'}$ of $H/H'$ input/output channels:

$$f_\sigma(\boldsymbol{x}) = \sigma(\boldsymbol{x}\Lambda + \mathbf{1}\mathbf{1}^\intercal \boldsymbol{x}\Gamma), \qquad \mathbf{1} = [1, \cdots, 1] \in \mathbb{R}^K, \qquad \Lambda, \Gamma \in \mathbb{R}^{D \times D'}$$

and $\sigma(\cdot)$ an element wise non-linear function. We stack two of those, one $f_{\text{relu}} : \mathbb{R}^{K \times H_x} \to \mathbb{R}^{K \times H_x'}$ with $\sigma()$ being the $\text{relu}(x) = \max(0, x)$ and a second $f_{\text{linear}} : \mathbb{R}^{K \times H_x'} \to \mathbb{R}^{K \times 1}$ without any non-linearity $\sigma()$. This structure on top of preserving the desirable permutation equivariance property it also brings a significant reduction of parameters since an increase of the number of users does not necessitate additional parameters with bigger network prone to overfitting traps.

Before the final non-linearity, which is a smooth approximation of ReLU, namely $\text{softplus}(x) = \log(1 + e^x)$, guaranteeing the output is positive, there is a critical normalization step $\boldsymbol{x} \to \frac{\boldsymbol{x} - \mathbb{E}[\boldsymbol{x}]}{||\boldsymbol{x}||_2}$ with $|| \cdot ||_2$ being the $\ell_2$ norm. To better understand the criticality of that step, consider the case of full-CSI where the output denotes how valuable each user is. Without the normalization step, the value network perceives that the higher the value assigned to a user, the more probable is to get resources, be satisfied and take reward, leading to a pointless attempt of increasing every user's value. However, by subtracting the mean value, whenever the value of a user increases, the value of the rest decreases, giving hence the sense of the total resources being limited. In the case of no-CSI, there is an additional benefit. Here there is an extra final operation, i.e., $\boldsymbol{x} \to \frac{\boldsymbol{x}}{||\boldsymbol{x}||_1}$, see Figure 1, so as to signify portions (of the total bandwidth) adding up to 1. Having done the normalization step previously (dividing by $||\boldsymbol{x}||_2$), helps keeping the denominator $||\boldsymbol{x}||_1$ stable.

A final note regards the exploration. The output has to satisfy properties (like positivity and/or adding to 1) which makes the common approach of adding noise on the actions cumbersome. An easy way out is through noisy networks (Fortunato et al., 2018), which introduce noise to the weights of a layer, resulting to changed decision of the policy network. The original approach considers the variance of the noise to be learnable; we keep it constant though since it provides better results. The noise is added at $\phi_{users}$ parameters, resulting to altered output features per user and therefore different allocations.

### 3.2.2 VALUE NETWORK

As mentioned previously $Q^{\pi_\theta}(s, a)$ is used for computing the gradient (3.2.1), however as it is intractable to compute, a neural network, called value network, is employed to approximate it. The common approach is through the Bellman operator

$$\mathcal{T}^\pi Q(s_t, a_t) = \mathbb{E}[R(s_t, a_t)] + \gamma \mathbb{E}_{s_{t+1} \sim P(s_t, a_t)}[Q(s_{t+1}, \pi(a_t))]$$

to minimize the temporal difference error, i.e., the difference between before and after applying the Bellman operator. This leads to the minimization of the loss

$$\mathcal{L}_2(\psi) = \mathbb{E}_{s_{t_0} \sim \text{P}_{t_0}, s \sim \rho_{s_{t_0}}^{\pi_\theta}}[(Q_\psi(s, a) - \mathcal{T}^{\pi_{\theta'}} Q_{\psi'}(s, a))^2]$$

where $(\pi_{\theta'}, Q_{\psi'})$ correspond to two separate networks called target policy and target value networks, respectively, used for stabilizing the learning, and are periodically (or gradually) updated as copies of the current actor and value networks.

Another approach is the following: instead of only approximating the expected value of the return, we approximate its distribution as in (Barth-Maron et al., 2018). Algorithmically, it is impossible to

represent the full space of probability distributions with a finite number of parameters so the value neural network $\mathcal{Z}_\psi^{\pi_\theta} : \mathcal{S} \times \mathcal{A} \to \mathbb{R}^{N_Q}$ must approximate with a discrete representation the actual $Z^{\pi_\theta}$. Among other variations (Bellemare et al., 2017; Dabney et al., 2018a) one can choose the representation to be a uniform (discrete) probability distribution supported at $\{(\mathcal{Z}_\psi^{\pi_\theta})_i, i \in \{1, \cdots, N_Q\}\}$ where $(\mathcal{Z}_\psi^{\pi_\theta})_i$ is the $i$-th element of the output. More rigorously, the distribution that the value neural network represents, is $\frac{1}{N_Q} \sum_{i=1}^{N_Q} \delta_{(\mathcal{Z}_\psi^{\pi_\theta})_i}$ where $\delta_x$ is a Dirac delta function at $x$ (Dabney et al., 2018b). Minimizing the 1-Wasserstein distance between this distribution and the actual one of $Z^{\pi_\theta}$ can be done by minimizing the quantile regression loss

$$\mathcal{L}_1(\psi) = \sum_{i=1}^{N_Q} \mathbb{E}_{s_{t_0} \sim \mathrm{p}_{t_0}, s \sim \rho_{s_{t_0}}^{\pi_\theta}, Z \sim \mathcal{T}^{\pi_{\theta'}} \mathcal{Z}_{\psi'}^{\pi_{\theta'}}(s_t, a_t)} [f_i(Z - (\mathcal{Z}_\psi^{\pi_\theta})_i)]$$

where $f_i(x) = x(\frac{2i-1}{2N_Q} - \mathbf{1}_{\{x<0\}})$ with $\mathbf{1}$ being the indicator function, the distributional Bellman operator is $\mathcal{T}^\pi Z^\pi(s_t, a_t) \overset{D}{=} R(s_t, a_t) + \gamma Z^\pi(s_{t+1}, \pi(a_t)), s_{t+1} \sim P(s_t, a_t)$ and $\mathcal{Z}_{\psi'}^{\pi_\theta}$ being the target policy network (defined in the same way as before).

An important observation is that even though we approximate the distribution of $Z^{\pi_\theta}(s, a)$, what we need at then end is only its expected value, approximated as $Q^{\pi_\theta}(s, a) \approx \frac{1}{N_Q} \sum_{i=1}^{N_Q} (\mathcal{Z}_\psi^{\pi_\theta})_i$. Therefore a natural question that arises here is why using $\mathcal{Z}_\psi^{\pi_\theta}$ instead of a simpler $Q_\psi(s, a)$ approximates straight away the needed expected value. If instead of having a scheduler and its users, we consider a teacher and its students, things become evident. Even though the objective of the teacher is to increase the mean "knowledge" of its students, using the distribution of the capacity/knowledge of the students enable for example deciding whether to distribute its attention uniformly among students or focus more on a fraction of them.

Even though intuitively we expected to observe gains by using the distribution, that was not the case at first. The main problem was that the distribution $Z^{\pi_\theta}$ was far away from 0 making it very difficult for for the policy network to well approximate them. One way to solve this could have been through scaling the rewards (i.e., the rewards are divided by the standard deviation of a rolling discounted sum of rewards). We came out with a new proposal and we propose to center the distribution through a *dueling* architecture (Wang et al., 2016). As shown in Figure 1 just before the output there is the dueling architecture with:

$$(\mathcal{Z}_\psi^{\pi_\theta})_i = \mathcal{Z}_\psi^{\pi_\theta, Mean} + (\mathcal{Z}_\psi^{\pi_\theta, Shape})_i - \frac{1}{N_Q} \sum_{i=1}^{N_Q} (\mathcal{Z}_\psi^{\pi_\theta, Shape})_i, \forall i \in \{1, \cdots, N_q\}$$

which effectively pushes $\mathcal{Z}_\psi^{\pi_\theta, Mean}$ to approximate $Q^{\pi_\theta}$ used for training the policy. To further encourage the decomposition of the shape and the center of the distribution, we add a loss term $L_{shape} = ||\mathcal{Z}_\psi^{\pi_\theta, Shape}||_2$, centering $\mathcal{Z}_\psi^{\pi_\theta, Shape}$ around zero.

In Figure 2 we provide additional element to support the choice of distributional reinforcement learning. We use the traffic model described in Table 1a showing the two classes of users with different requirements. In Figure 2a (which is the mean taken over five experiments) we see that all approaches finally converge to approximately the same value; nevertheless, the combination of distributional with dueling is faster. Figures 2b and 2c focus on two (out of the five) experiments, where it is evident the advantage of the distributional approach. This approach is able to detect the existence of two difference classes with different requirements, thus gradually improving the satisfaction rate for both of them. On the other hand, trying only to learn the expected value leads to a training where one class is improved at the expense of the other.

A final remark is that the architecture should be designed in a way preserving the permutation invariance. If we associate every user's characteristics with the resources given by the agent, i.e., the action corresponding to it, then permuting the users and accordingly the allocation should not influence the assessment of the success of the agent. To build such an architecture, we adopt the one of the Policy Network using ideas from (Zaheer et al., 2017).

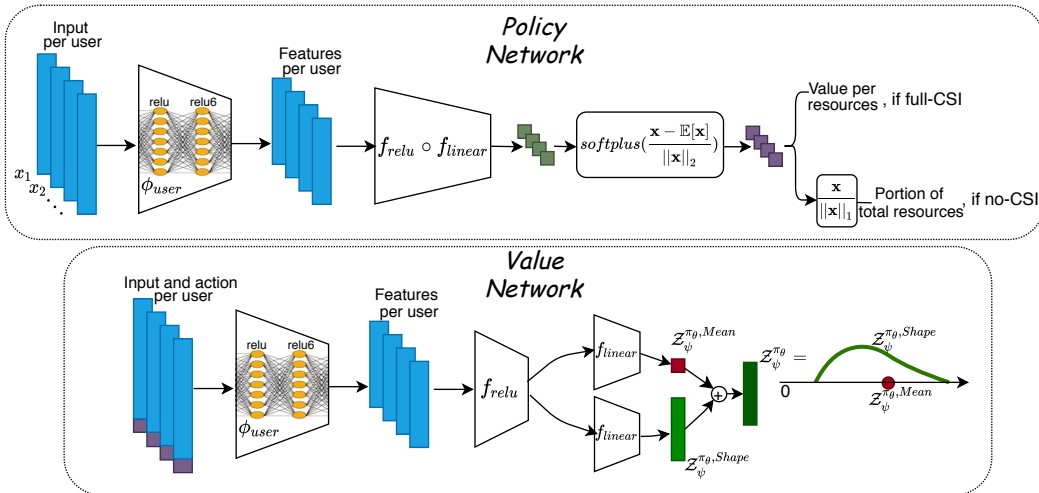

Figure 1: The Neural Network architecture.

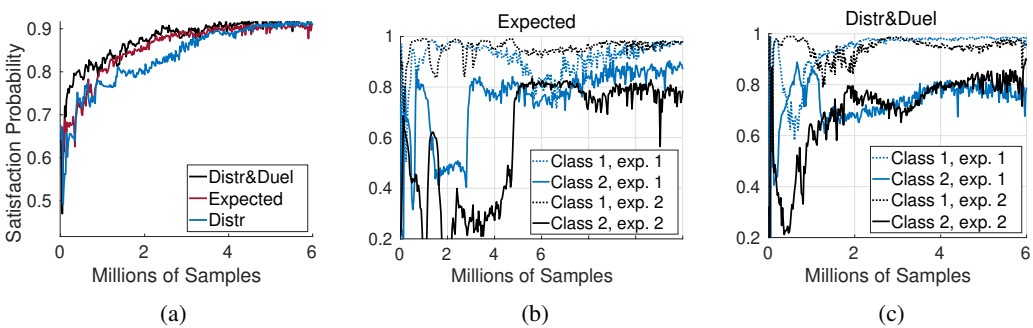

Figure 2: Comparison between distributional and traditional (non-distributional) approach. We conducted five experiments (with different seeds) for no-CSI using the traffic model of table 1a and a maximum number of users $K = 75$. In the first figure we depict the average over those five experiments; in the other figures we consider two specific experiments (named exp. 1 and exp. 2).

## 3.3 BENCHMARKS USING CONVENTIONAL APPROACHES

### 3.3.1 FULL-CSI

For explanation convenience, we use once again the analogy introduced in section 3.1. Having full-CSI means that the server knows for all people (active users) at each time, the size of their glasses and their importance. If we try to solve the problem myopically ignoring the impact on the future steps, we have a reformulated knapsack problem with the size of the glasses being the weight of the objects, whose value is the importance of its "holder". The size of the server's pitcher is the capacity of the knapsack that we try to fill with objects so as to maximize the sum of their value. We refer to this benchmark simply by *myopic Knapsack*. More details are given in appendix B.2.1.

Accounting for the impact on the future is not trivial (discussed in appendix B.2.2). One way is to assume that the scheduler acts as an oracle and knows in advance for the future $T - 1$ time steps which users will appear and what will be their channel qualities. In that case, the problem can be written as an Integer Linear Programming and can be solved by standard Branch and Cut approach. We obtain therefore an upper bound which we call *oracle ILP*. More details in appendix B.2.2.

Table 1: Classses description for two scenarios

(a) Equal importance users

|  | Data | Lat. | Imp. | Prob. |
|---|---|---|---|---|
| class 1 | 256 Bytes | 2 | 1 | 0.3 |
| class 2 | 2048 Bytes | 10 | 1 | 0.2 |

(b) Prioritized and normal users

|  | Data | Lat. | Imp. | Prob. |
|---|---|---|---|---|
| class 1 | 256 Bytes | 2 | 1 | 0.15 |
| class 1+ | 256 Bytes | 2 | 2 | 0.05 |
| class 2 | 2048 Bytes | 10 | 1 | 0.3 |
| class 2+ | 2048 Bytes | 10 | 2 | 0.05 |

### 3.3.2 NO-CSI

Without CSI, one cannot do much so one may want at least to learn (usually with autoregression) the channel model. Using a DRL approach, the dynamics of the environment do not change through the training (and testing) and the agent learns how to react well under those conditions. Therefore, even though it acts in a model-free manner, in a way it learns the statistics of the problem. For fair comparison, we consider a benchmark that knows the statistics of the channel and traffic dynamics. Based on knowing the distributions of the problem, one can reach to an optimization problem with multiple local optimums. The Frank-Wolfe algorithm guarantees reaching to a local optimum, so we run this method $N_{init}$ times and pick the best found local optimum. More details in appendix B.1.

## 4 EXPERIMENTS

We consider two cases for the traffic, described in table 1. The first one has two classes, one with requiring a low amount of data but with a stringent latency constraint (of just two time slots) and another one with the opposite. Every class has the same importance which is the main difference with the second scenario where a certain part of users are of highest priority. An important remark is that when users are all of equal importance, then the objective of the scheduler coincides with indiscriminately increasing the satisfaction probability of every user. Finally the Prob. column describes the probability a user of the class to appear at a time slot (note that they do not add up to one since it is possible that no user appears).

The channel dynamics is described through a parameter $\rho$. For $\rho = 0$ the dynamics behaves in an i.i.d. manner increasing the unpredictability of the future but also the chances to recover from a bad quality connection. We consider users appear with distance from the base station varying from 0.05 km to 1 km. We keep the energy per symbol(/bandwidth) equal to $1\mu J$. For more information on the architecture of the networks and the setting, the interested reader is referred to appendix C. Hereafter we refer to the DRL approach as "Deep Scheduler".

In Figures 3a,3b we demonstrate the high performance of the Deep Scheduler for the full CSI case with a maximum number of $K = 100$ users. It consistently manages to outperform the myopic Knapsack approach, which is myopically optimal. We emphasize that solving a Knapsack problem is non-polynomial. We also show that actually the performance of Deep Scheduler is close to optimal since it is close to the "oracle ILP", which is an upper bound since the scheduler knows in advance the future.

In Figures 3c and 3d we focus on the no-CSI case with $K = 60$ users. In that case, we know that the Frank Wolfe (FW) method reaches to a suboptimal solution. We repeat the algorithm for many initialization (for each time step) to get the best possible solution among suboptimal ones. Note that this process is unfortunately very slow (see appendix B.1) and gets slower for increasing $K$. So even if the method shows considerable improvements with higher $N_{init}$, we were obliged to stop at 20. Moreover, as $K$ increases, unfortunately so does the number of local optimums and and that of solutions with poor performance. This is why we see the Deep Scheduler to substantially outperform the FW for even a moderate $K = 60$.

Finally we include Figure 3e, which showcases that the Deep Scheduler consistently keeps its high performance even if the traffic model gets more complicated so as to represent more accurately a real-world scenario.

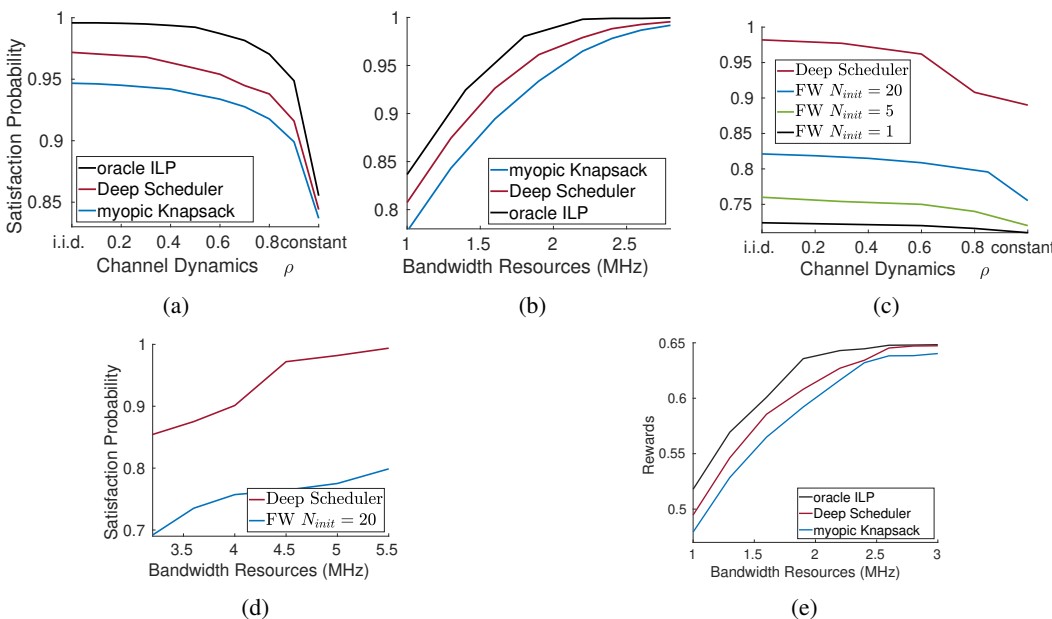

Figure 3: Performance of Deep Scheduler versus the benchmarks on different scenarios. The first four correspond to the case of Table 1a) and the last to the other. Figures 3a,3b,3b refer to the case of full-CSI and the rest to no-CSI.

## 5 CONCLUSION

The problem of scheduling and resource allocation for a time varying set of clients with heterogeneous traffic and QoS requirements in wireless networks has been investigated here. Leveraging on deep reinforcement learning, we have proposed a deep deterministic policy gradient algorithm, which builds upon distributional reinforcement learning and Deep Sets. Experimental evaluation of our proposed method in scenarios with different traffic and wireless channel dynamics, shows significant gains against state-of-the-art conventional combinatorial optimization methods.

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
