# OpenReview forum: "Deep Reinforcement Learning For Wireless Scheduling with Multiclass Services"
_ICLR.cc/2021/Conference — Reject_

### Official Review · AnonReviewer1 · 2020-10-26
**Interesting application but has clarity issues and perhaps not a good fit for ICLR**

**Rating:** 3
**Confidence:** 3

**Review:**

The authors propose a deep RL solution for the communication problem of user scheduling and resource allocation. The deep RL solution uses deterministic policy gradient + quantile regression + dueling + deep sets and the authors demonstrate it outperforms classical solutions on benchmark tasks.

Recommendation:

I quite frankly have no prior knowledge of the task this paper aims to solve. It didn't come across as a grandstanding AI challenge in any capacity (feel free to debate this) so I'm looking at this paper from the angle of significance to the deep RL community. In general, the questions I'm asking are: Does this paper introduce a novel deep RL algorithm? Is the knowledge produced by this paper generalizable to other problems or algorithms? Does this paper provide value to anyone who isn't concerned with the specific application?

As of now, I feel like the answer is no to all three and so I would recommend rejection to this conference. However, there are presumably venues which are a better fit and I hope the authors consider submitting elsewhere.

There were other issues with the paper. In general, clarity and organization were a big issue for me. The authors were very rigorous with their supplementary material, so I believe most of the information is there, but was not presented in an easily digestible manner.

Strengths:
- Thorough supplementary material and code is provided.
- The use of deep RL to tackle the problem felt well-motived and a good fit.
- The performance of the agent seems strong but I'm not clear on the significance of some of the results.

Weaknesses:
- I think the problem set up was clear in the sense that I understood the overarching objective. However, the specifics of the problem, specifically in the context of RL was not. The paragraph about 3.2 is a generic description of the RL problem and left me wondering the connection to the actual application. I realize many of the details are contained in the supplementary material but the statement "the problem can be modeled as an MDP" was not defended & the following description did not clarify the problem statement. For example, immediately after, in 3.2, "high variance randomness" is discussed but its not clear to me why this is the case or how this randomness affects the problem- reward? transitions?
- The structure of the paper did not feel helpful to me. 3.2 is categorized into "Policy Network" and "Value Network" for somebody who is comfortable with RL a lot of the details felt unnecessary but more importantly, this organization doesn't provide a solid presentation of the algorithm. Somebody who is interested in the application and is not an RL expert, won't necessarily follow why "Policy network" is being presented or what that even means necessarily. There isn't a clear overview of the algorithm.
- Novelty of the algorithm is low in the sense that it is a combination of prior, existing ideas.
- I felt like many of the algorithmic choices were not well-justified. For example, the use of QR is justified by an analogy? The use of the dueling architecture also seems unusual when the authors also propose a much simpler solution. It was also unclear what the issue with "The main problem was that the distribution Z was far away from 0 making it very difficult for the policy network to well approximate them" exactly meant.

Minor Comments:
- There are a few latex issues with reversed quotations.
- "In Figure 2 we provide additional element to support the choice" -> an additional
- The objective of Figure 2 is nice but its confusing to have two sets of experiments presented in the same graph. Class is not explained in the description of the graph and the significance of the graph is not explained in the figure description.

**Post-Rebuttal

I appreciate the authors taking the time to respond. Unfortunately, my belief that this paper is not strong enough from the deep RL perspective to warrant acceptance has not changed. If other combinations of tools do not work, then the authors should improve this justification, with ablation studies or stronger theoretical motivation. I’ll add that my score is not influenced by my concern that this paper may not be a good fit for ICLR (I leave that choice to the AC).

---

> ### Author Response · Authors · 2020-11-25
> **Justifying the applicability of our paper to ICLR**
>
> We would like to thank the Reviewer for his/her nice remarks. Indeed, we believe that the code should be available (we passed it now to an anonymoys repository) in order to be able the reproduce the results.
> We are also pleased to know that you agree with our motivation on using/applying a DRL approach to this problem.
>
> First, we would first to point out that it is an actual challenge to build such an agent. We have recently came through similar open challenges/competitions, such as the one of NOKIA https://github.com/nokia/wireless-suite/. The first problem "TimeFreqResourceAllocation-v0" is very close to our set-up and our ideas could be adapted (in a future work) to their setting. Additionally a scheduling problem with multiple classes is very relevant in many aspects. You can think of computing resources of a company (e.g. Amazon, Microsoft) where they have plenty of users (and some of them prime) needing different type of service requirements and resources. Maybe some of them belong to the class of many CPUs and other of GPUs. How to allocate them is challenging. Also other scenarios coming from finance (stocks) and smart grid could easily show resemblance to our setup. Therefore we believe we can adjust and apply our algorithms to these fields, expecting to obtain promising performance.
>
> As far as the novelty is concerned we believe that combining tools that have already been proposed in a novel way (for example using the dueling architecture to bring the shape of the distributions to the zero and more easily estimate them) is a novelty. The problem was very hard and a random combination of tools doesn't work. Also we  would like to emphasize that the call for papers ICLR explicitly says that relevant topics are "applications in audio, speech, robotics, neuroscience, computational biology, or any other field" aare welcome and we believe that our paper is relevant and fits to this venue.
>
> Passing to the other mentioned weaknesses, we rewrote the paper with a dedicated subsection describing the MDP problem where we also explain also the randomness coming from the traffic and channel dynamics. Also we hope that the section now where the policy network is explained is more clear. As we also mention to the first reviewer "we make a one analogy with teacher-students in order to just  build intuition why it might be helpful. We are fully aware though that a simplifying analogy is not enough. This is why we provided Figure 2 (Figure 3 in the revised paper). We improved the presentation (also by making the Fig 3b&3c less dense) so now it is easier to read the plots. Seeing the figure 3a we show a curve [expected] which corresponds to disregarding the distributional perspective. By using Distr. RL and adding some tricks (Dueling & Reward Scaling) it outperforms the expected. To provide further explanation on why we opt for distributional RL we provide the Fig3b&3c which shows how expected and distributional are handling the different classes. In contrast to Expected, distributional RL understands quite fast the existence of two different classes of users and steadily improves both of them."
>
> Finally it was right your idea to add the normalization of rewards. But we emphasize that the reward scaling is doing some different operation. Please see the figure  1 where is nicely depicted. We show in our experiments (Figure 3a) that it helps to have both dueling and reward scaling.

---

### Official Review · AnonReviewer3 · 2020-10-28
**This paper investigates the deep reinforcement learning method to schedule the traffics with heterogeneous requirements under dynamic channel environments. The proposed method is compared with its lower and upper bound methods, i.e., so called myopic Knapsack and oracle ILP. It is clear that the proposed method has the merits compared with the lower and upper bound methods in terms of performance and implementation.**

**Rating:** 7
**Confidence:** 3

**Review:**

Basically, it seems that the proposed method is interesting and meaningful. The scheduling problem in this paper is based on the analogy to a server having a water pitcher, and the deep reinforcement learning approach for the scheduling problem has been designed. However, the scheduling problem in wireless networks is a very famous issue. Of course, applying DRF to it is quite interesting. However, the authors need to describe the conventional well-known scheduling algorithms and compare them with the proposed scheme (now, the current paper only focuses on applying the DRF to the scheduling and evaluating its performance in aspects of an optimization problem.). Further,  typically, in scheduling problems, efficiency (total data rate) and fairness are the key factors and it is needed to describe the relationship between these conventional performance metrics and the satisfaction probability.

---

> ### Author Response · Authors · 2020-11-25
> **We an environment using real-data and we compare also under total data rate metric.**
>
> We are pleased to listen that you enjoyed our paper and also the interpretations that we gave.
> Due to space limitation we could only briefly describe the conventional approaches in the main paper and we give more details in the appendixes.
>
> Further we agree that we have to also compare the algorithms  with metric being the total data rate. We added a new section (4.1) in which we built a similar environment that uses data coming from real world measurements. We also added a new powerful scheduling rule called exponential Rule against which we compare. In that environment we show the performance also using the total data rate. The fairness metric is actually a bit tricky. Firstly because every user has different needs so giving to all of them a fair amount of data is not actually desirable. Some of them they just don't need them. Also by construction we assume some of the classes are more important so as to account for users with more expensive contracts. At this point it actually desirable to be unfair and provide them better service.

---

### Official Review · AnonReviewer2 · 2020-10-28
**Promising Application of DRL to a classic wireless problem**

**Rating:** 7
**Confidence:** 4

**Review:**

This paper addresses the long standing problem of scheduling and resource allocation in wireless networks using modern Deep Reinforcement Learning techniques.
It is clearly written and easy to follow but suffers from several minor typos.
The methodology is well justified and thoroughly motivated.
Experimental evaluation seems thorough and provides convincing results.

The MDP is not described thoroughly enough:
What is your reward, action space, state space, observations?
How is the allocation deadline incorporated into the reward?
It would be nice to have these details listed in a sub-section somewhere in Section 3.

Regarding the evaluation, "synthetic" traffic patterns are used.
Can you use real world traces with a simulator for evaluation (similar to https://github.com/hongzimao/pensieve)?
Also real world applicability is not addressed?
Will the inference times for the deep network lead to any significant overheads when measured at the time scale of wireless communications?
Overall, the evaluation setup seems preliminary to me and needs more work to provide assurance of real world usability.

---

> ### Author Response · Authors · 2020-11-25
> **We thank the reviewer for giving us access to real data**
>
> We would like to thank you for your positive review. Indeed the MDP problem was not clearly stated so we added the section 3.1.1. that now clarifies it explicitly describing the state, action and rewards.The deadline is incorporated because if a user is not satisfied within his latency constraint then it doesn't belong to set of active users $U_t^{active}$ and so the reward that could possibly the agent take from it is lost.
> To address the applicability issue and try to justify the possibility of such a solution being implemented in reality we used some of the traces that you provided, specifically the ones from Belgium where they used wireless networks (LTE/4G). We needed again to add some assumptions that we explain in section 4.1 but since we got high performance we are more optimistic for the real-world applicability.

---

### Official Review · AnonReviewer4 · 2020-10-30
**An interesting application of DRL, but the paper could be improved.**

**Rating:** 5
**Confidence:** 3

**Review:**

Paper Summary
This paper investigated the problem of scheduling and resource allocation for a time-varying set of clients with heterogeneous traffic and QoS requirements in wireless networks. It proposed to solve this problem with distributional based DDPG with Deep Sets, and conducted experiments showing performance gains against conventional methods.

Paper Strength
1.	The paper considered a complex scheduling scenario, which is a hard problem by conventional optimization methods. The problem setting takes into account traffic model, geometry model, channel model, and rate model. Both full-CSI and partial-CSI scenarios are considered.
2.	The paper adopted state-of-the-art techniques and works fine. Specifically, Distributional RL and Deep Sets for speeding up the convergence and reducing neural network parameters, respectively.
3.	The proposed algorithm outperforms conventional combinatorial optimization methods.

Paper Weakness
1.	The presentation of the paper should be improved. Right now all the model details are placed in the appendix. This can cause confusion for readers reading the main text.
2.	The necessity of using techniques includes Distributional RL and Deep Sets should be explained more thoroughly. From this paper, the illustration of Distributional RL lacks clarity.
3.	The details of state representation are not explained clear. For an end-to-end method like DRL, it is crucial for state representation for training a good agent, as for network architecture.
4.	The experiments are not comprehensive for validating that this algorithm works well in a wide range of scenarios. The efficiency, especially the time efficiency of the proposed algorithm, is not shown. Moreover, other DRL benchmarks, e.g., TD3 and DQN, should also be compared with.
5.	There are typos and grammar errors.

Detailed Comments
1.	Section 3.1, first paragraph, quotation mark error for "importance".
2.	Appendix A.2 does not illustrate the state space representation of the environment clearly.
3.	The authors should state clearly as to why the complete state history is enough to reduce POMDP for the no-CSI case.
4.	Section 3.2.1: The first expression for $J(\theta)$ is incorrect, which should be $Q(s_{t_0},\pi_\theta(s_{t_0}))$.
5.	The paper did not explain Figure 2 clearly. In particular, what does the curve with the label "Expected" in Fig. 2(a) stand for? Not to mention there are multiple misleading curves in Fig. 2(b)&(c). The benefit of introducing distributional RL is not clearly explained.
6.	In Table 1, only 4 classes of users are considered in the experiment sections, which might not be in accordance with practical situations, where there can be more classes of users in the real system and more user numbers.
7.	In the experiment sections, the paper only showed the Satisfaction Probability of the proposed method is larger than conventional methods. The algorithm complexity, especially the time complexity of the proposed method in an ultra multi-user scenario, is not shown.
8.	There is a large literature on wireless scheduling with latency guarantees from the networking community, e.g., Sigcomm, INFOCOM, Sigmetrics. Representative results there should also be discussed and compared with.

======
post rebuttal: My concern regarding the experiments remains. I will keep my score unchanged.

---

> ### Author Response · Authors · 2020-11-25
> **We improved the presentation and rewrote the paper with new experiments**
>
> We agree that the problem we are tackling in this paper is challenging and hard to solve. Indeed, it was not easy to come up with an architecture and an algorithm that manages to converge to a point that exhibits very good performance. Moreover, we are targeting practically relevant scalable solutions. Our main problem was that scaling up to a large number of users (e.g., 100) substantially increases the stochasticity of the environment, resulting in various issues, including convergence problems. On top of that, we wanted to compare with and outperform strong baselines. We were finally able to come up with an architecture combining Deep Sets followed by a normalization that could show significantly better performance against the conventional methods. Finally using a distributional type of RL, we have observed consistency in training with a gradual improvement of the performance in the training.
>
> Concerning the paper weaknesses:
> 1)Indeed, the presentation of the paper was confusing in some points and as requested by other reviewers, several clarifications in the problem description were required. We created a subsection called "MDP formulation and state representation" where we briefly yet clearly descibe the problem considered here, the traffic generation model, its MDP formulation and what are the states-actions-rewards. That way, we passed information previously given in the appendix into the main text.
>
> 2)For the DeepSets, in the paragraph where we explain them, we mention that not only they preserve the permutation equivariance which is an inherent property of our problem (i.e. swapping two users should lead to the corresponding swap of their allocations) but also brings a huge parameter reduction (thus, less prone to overfitting) since no parameter increase is required when scaling up the number of users.
>
> For the Distributional RL, we make at first one analogy with teacher-students in order to build intuition why it might be helpful. We are fully aware though that a simplifying analogy is not enough. This is why we provided Figure 2 (Figure 3 in the revised paper). We improved the presentation (also by making the Fig 3b&3c less dense) so now it is easier to read the plots. In figure 3a we show a curve [expected] which corresponds to disregarding the distributional perspective. By using Distr. RL and adding some tricks (Dueling & Reward Scaling) it outperforms the expected. To provide further explanation on why we opt for distributional RL we provide the Fig3b&3c which shows how expected and distributional are handling the different classes. In contrast to Expected, distributional RL understands quite fast the existence of two different classes of users and steadily improves both of them.
>
> 3)As mentioned previously we state them now clearly in a dedicated subsection.
>
> 4)Now that the action space is described in a clearer manner and it is easier to see why it is impossible to apply DQN. The action space for  full-CSI is $2^{Number Of Users}$ and for no-CSI it is continuous which, as we mention in the main text, prohibits the use of DQN. As suggested, we also implemented the TD3 version, tested on the environment and using real data (section 4.1). However, TD3 showed worse results than the Distributional RL approach and it has been omitted. Nonetheless we would like to point out that our main scope is to show gains against conventional approaches so we did not want to open up too much on the plenty approaches for DRL existing in literature. Finally how time efficient the algorithm is in terms of training is shown Figure 3 where it is shown that it needs around 3 million samples to reach from zero to top performance. In terms of testing, we only used around one thousand parameters for the synthetic data environment and around four thousand for the real data experiments. In contrast, the other conventional approaches (except the exponential rule) need exponential running time.
>
> 5)We significantly revised the paper, correcting as well grammatical errors and typos.
>
> Detailed Comments:
> We replied to most of them except for
> 1.6) It is one of our future goals to also try to increase even more the number of classes. But we mostly used this setup driving from direction of 5G to add 3 classes with one being URLLC classes (Ultra reliable low latency Communications: short packets with stringent latency constraints and high reliablity) and eMBB (enhanced Mobile BroadBand : only goal is high data rates)
> 1.8)For wireless communication on the PHY - MAC layer the only paper that we know addressing the multi -class problem was the exponential rule against which we compare under an environment using real measurements.

---

### Public Comment · ~Rahif_Kassab1 · 2020-11-13
**Wireless communications paper?**

The idea/Application of DRL for wireless scheduling has been investigated extensively in many papers in the wireless community. A simple search on google scholar shows lots papers doing so and some of them specifically using DDPG, MADDPG, DQN ...(for e.g., https://arxiv.org/pdf/2009.08346.pdf ; https://ieeexplore.ieee.org/stamp/stamp.jsp?tp=&arnumber=8896945; https://arxiv.org/pdf/1905.05914.pdf ; https://ieeexplore.ieee.org/stamp/stamp.jsp?tp=&arnumber=8757174 ...)

That being said, can the authors explain how is their technique novel from an "ML" point of view and not from a "wireless communications" point of view? For instance, what are the new ML features of your algorithm?

On another note, for what type of wireless services/classes does your model apply to ?
Thanks.

---

> ### Author Response · Authors · 2020-11-25
> **Answers to your concerns**
>
> We would like to thank you for your questions. We knew the publications that you mentioned but none of them does a centralized scheduling with a traffic of users belonging to diverse classes so we believe that are not closely related.  Your concern about combining only existing tools is already pointed out by the last reviewer where we reply.
> Finally you can think the classes being users from URLLC (ultra reliable low latency) who need short packet but with the minimum possible delay and eMBB (enhanced Mobile BroadBand ) where high data rates are desired without the need for strict latency constraints.

---

### Decision · Program_Chairs · 2021-01-07
**Final Decision**

**Decision:**

Reject

**Comment:**

The reviewers mostly agree that this paper presents a new deep reinforcement learning-based approach to solving a challenging problem in the communications domain -- wireless scheduling. However, the main concern, expressed almost unanimously, is about the novelty of the ideas in the paper beyond the assembly of existing deep RL techniques and the translation of the scheduling problem to the language of MDPs in a careful manner that respects modern communication systems standards such as 5G (e.g., URLLC and eMBB traffic demands). A secondary concern, also expressed during the author rebuttal discussion, is about adequate comparison to competing approaches motivated from the literature in wireless scheduling. In view of these issues, I suggest that the author(s) explore more appropriate avenues to submit this piece of valuable translational work, including venues that address the specific topic of wireless communication where a more comprehensive evaluation and comparison could be possible.

(NOTE: The comments and evaluation above disregard the "enhanced" draft submitted by the author(s) during the rebuttal phase. I was informed that the submission was reverted to the original draft due to space constraints being exceeded in the enhanced version.)